# Methodology for Evaluating the Generalization of ResNet

**Anan Du** [1,2], **Qing Zhou** [1,*] and **Yuqi Dai** [1,2]

1   National Space Science Center, Chinese Academy of Sciences, Beijing 100190, China;
    duanan21@mails.ucas.ac.cn (A.D.); daiyuqi18@mails.ucas.ac.cn (Y.D.)
2   University of Chinese Academy of Sciences, Beijing 100049, China
*   Correspondence: zhouqing@nssc.ac.cn

**Abstract:** Convolutional neural networks (CNNs) have achieved promising results in many tasks, and evaluating the model's generalization ability based on the trained model and training data is paramount for practical applications. Although many measures for evaluating the generalization of CNN models have been proposed, the existing works are limited to small-scale or simplified model sets, which would result in poor accuracy and applicability of the derived methods. This study addresses these limitations by leveraging ResNet models as a case study to evaluate the model's generalization ability. We utilized Intersection over Union (IoU) as a method to quantify the ratio of task-relevant features to assess model generalization. Class activation maps (CAMs) were used as a representation of the distribution of features learned by the model. To systematically investigate the generalization ability, we constructed a diverse model set based on the ResNet architecture. A total of 2000 CNN models were trained on the ImageNet subset by systematically changing commonly used hyperparameters. The results of our experiments revealed a strong correlation between the IoU-based evaluation method and the model's generalization performance (Pearson correlation coefficient more than 0.8). We also performed extensive experiments to demonstrate the feasibility and robustness of the evaluation methods.

**Keywords:** generalization evaluation; ResNet; convolutional neural network (CNN); class activation map (CAM)

## 1. Introduction

Convolutional neural networks (CNNs) are renowned for their exceptional data representation and fitting capabilities, typically exhibiting robust generalization performance with unseen data. In real-world applications such as autonomous driving and the medical field, it is crucial to evaluate the model's generalization ability based on the trained model and its training data. The generalization ability of a model is defined as the accuracy of the model using unseen data under independent identically distributed (i.i.d.) conditions [1]. However, the generalization of over-parameterized CNN models is not well understood theoretically, and there is a lack of precise methods to accurately evaluate this ability. This leaves the assessment of CNN model generalization as a complex and unresolved issue, particularly due to the risk of overfitting, where the number of parameters of deep CNN models often surpass the size of the training dataset [2].

The generalization ability of neural networks is vital for their practical application, thus necessitating a rigorous evaluation framework [3,4]. If $\mathbb{P}$ is the data distribution, the training data $\mathcal{D}_{train} = \{(\mathbf{x}_i, y_i)\}_{i=1}^N$ is a set of input–output pairs sampled i.i.d. from $\mathbb{P}$, where $\mathbf{x}_i$ represents a sample of data, $y_i$ represents the label of the sample, and N represents the number of training samples. The model is trained on a known training dataset. However, the test data $\mathcal{D}_{test}$ are unknown, which is also obtained by independently sampling from $\mathbb{P}$. Evaluating the model's performance on unseen data, which is its generalization ability, requires consideration of both the known training data and the trained model itself.

Generalization evaluation can be formally defined as $\mu(f, D_{train}) \rightarrow Acc_{D_{test}}(f)$, where $\mu$ is the evaluation method and $f$ represents the trained model.

A good assessment method should have a strong correlation with the generalization gap [3,5], a pivotal concept introduced in [1]. The generalization gap represents the model's generalization capability, and is the difference in accuracy between a model's training and test set performances upon convergence. A narrower generalization gap value indicates a better generalization ability of the model, while a wider gap suggests poorer generalization.

This paper delves into the generalization evaluation of CNN models, with a focus on ResNet for image classification, aiming to fill the existing gaps in knowledge. The methods for assessing the generalization of CNN models can be categorized into two main groups: theoretical analysis and empirical research. The former relies primarily on traditional statistical learning theory [6], encompassing concepts such as VC dimension [7] and margin theory in support vector machines. These theoretical approaches aim to establish an upper bound for the generalization error of deep neural network models [8,9]. However, these methods often lack experimental validation on extensive model sets and may not accurately provide the upper bounds for generalization errors [10]. The latter approach involves predictive modeling, where the generalization capability is forecasted based on the characteristics of network model weights [11] or intermediate layer outputs [1]. However, these methods are typically investigated on small-scale or simplified model sets [3], which limits their applicability and creates a gap between their findings and real-world tasks.

To overcome these limitations, we used IoU, a well-established metric in the field of object detection [12–15], as a method for evaluating the generalizability of CNN models, and constructed a model set to validate its effectiveness. The contributions of this paper can be summarized as follows:

- We provide a new perspective for evaluating the generalizability of CNN models by application of IoU. Specifically, we leverage IoU to compute the ratio of task-relevant features learned by the model as a measure for assessing generalizability. Task-relevant features are defined as features for the location of the category object in the image.
- A model set of residual network architecture [16] was meticulously constructed, including 2000 ResNet models trained on a subset of ImageNet [17]. This extensive ensemble was developed by methodically varying a set of standard hyperparameters.
- We conducted extensive experiments on the constructed model set and showed that using IoU as a criterion for evaluating the generalizability of CNN models is effective and robust.

## 2. Related Works

### 2.1. Generalization Evaluation Methods Based on Theoretical Analysis

Traditional statistical learning methods are extensively employed to evaluate the generalization performance of CNN models. Bartlett et al. [8] proposed an approach that utilizes the spectral norm product of the weight matrix as the Lipschitz constant for assessing the model's generalization ability. Another study [18] introduced a method that leverages the average weights of random models to enhance overall model generalization. Additionally, a separate study [9] delved into the generalization performance by formulating a loss function predicated on the marginal distribution. This approach specifically targets the optimization of the statistical properties inherent to the entire marginal distribution, with a particular emphasis on the ratio of marginal standard deviation to the expected margin. However, it is important to note that theoretical analyses often necessitate stringent assumptions, which may limit the applicability of the generalization evaluation methods. Nagarajan et al. [10] conducted experiments demonstrating the difficulty of accurately establishing an upper bound for model generalization ability using traditional theoretical methods. These findings highlight the need for more robust and flexible methodologies to accurately evaluate the generalization capabilities of CNN models.

### 2.2. Generalization Evaluation Methods Based on Modeling Prediction

Several empirical research studies [1,3,5,19,20] have proposed methods for predicting the generalization ability of neural networks. Jiang et al. [1] utilized quantile statistical values outputted by specific layers of the model to construct a regression model for predicting the model's generalization ability. They constructed small-scale and simplified model sets to train and validate generalization gap predictive models, such as linear and logarithmic models. A competition named Predicting Generalization in Deep Learning (PGDL) was proposed based on this model set to predict the generalization ability of deep neural network models. This competition promoted research on using experimental methods to build models to predict the generalization ability of neural network models. Chuang et al. [19] treated the middle layer of the model as a classifier and selected the output data of a specific layer to establish a prediction model based on margins for the model's generalization ability, employing optimal transmission theory. Schiff et al. [21] predicted the generalization ability of the model using perturbation response curves, which captured how a trained model's accuracy varies with different levels of perturbation in the input data and introduced two novel measures, the Gi-score and Pal-score, inspired by economic inequality metrics, to accurately predict generalization capability. These methods require not only training data but also labeled test data. Although these methods perform well on PGDL datasets, most of them require training data and have not been verified on large-scale models. Wei et al. [5] used the derivative of the loss function of the model parameters, that is, the Jacobian matrix, as a reference for evaluating the generalization ability of the model. However, this approach required significant computational resources. Deng et al. [22] used the model's invariance to transformations of the input data to evaluate generalization. Then, a more efficient method for generalizability assessment was proposed which uses the normalized value of the nuclear norm of the prediction matric to evaluate the generalization ability of a model [20].

Most of these empirical research works have been experimentally verified on simplified or small-scale model sets, which may lead to inaccurate or poor applicability of the obtained methods for evaluating model generalization. Jiang et al. [3] used correlation to analyze generalization evaluation methods on a large-scale model set and found that most methods have a weak correlation with the true generalization ability of the model.

In summary, the methods for assessing the generalization ability of deep neural network models can be categorized into theoretical analysis-based approaches and empirical research methods. Theoretical analysis methods, while widely used, often have limited applicability and struggle to provide accurate upper bounds for model generalization errors. Empirical research works have proposed various modeling prediction methods, but many of them rely on modeling and training and lack validation on large-scale and complicated model sets.

## 3. Method

### 3.1. Class Activation Map

Due to the black-box nature of deep neural network models, directly assessing their generalization ability is challenging. To address these challenges, gaining insights into the decision-making processes of these models can offer novel approaches for evaluating generalization. Various methods for explaining the decision-making of neural network models have been proposed. Among them, the Class Activation Map (CAM) based on the attribution explanation method is an efficient and accurate approach [23,24]. CAM can quantify the importance of different positions in the input image to the CNN model's final output. In other words, the class activation maps can be used to represent the feature weights learned by the model at different pixels in the input image. One study [25] suggested that irrelevant features learned by the network model will decrease the generalization ability of the model. Drawing from the principle of algorithm stability, it can be inferred that a model's generalization capability can be enhanced when it learns greater feature weights at positions that are pertinent to the task. Class activation map (CAM) [23,24] is a type of

attribution interpretation method that can calculate the correlation degree of each pixel in the image with the final prediction of the model.

As depicted in Figure 1, Grad-CAM [24], an evolution of CAM, is utilized to generate a heat map that graphically represents these correlations. The heat map's values are normalized to a scale where 0 signifies the least significant correlation, and 1 denotes the most substantial correlation. CAM was proposed in the field of image classification, especially for CNN models, which have good interpretability. It serves as a quantitative measure of the feature weights across various pixel positions within the image, as well as a representation of the spatial distribution of these learned features.

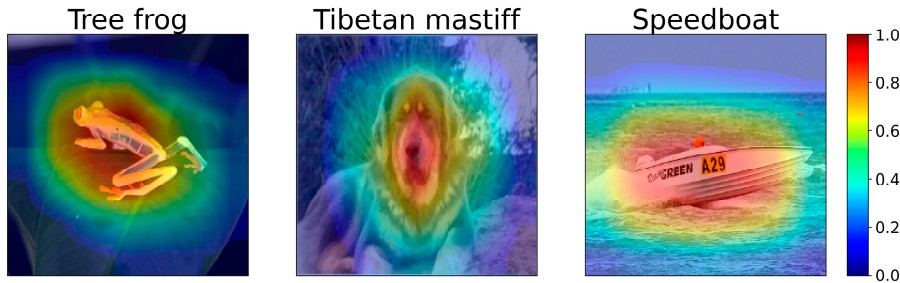

**Figure 1.** Examples of class activation maps.

For a given category c, the general calculation method of CAM is as follows:

$$M^c = \text{ReLU}(\sum_k w_k^c F_k) \tag{1}$$

where $M^c \in \mathbb{R}^{h \times w}$ represents the class activation map of the model for category c. $F_k \in \mathbb{R}^{h \times w}$ is the feature map output by the middle layer of the model, and $w_k^c$ represents the importance of the *k*-th dimension feature map relative to category c. The calculation method of CAM proposed by Zhou et al. [23] uses the weight parameter of the fully connected layer after the network convolution layer as $w_k^c$. This method requires adding a global average pooling layer (GAP), and class activation maps can only be calculated after the last layer of convolution. The Grad-CAM [24] method uses the gradient mean of the loss function on the feature map as $w_k^c$. This method can calculate class activation maps at any layer in the network without modifying the network structure.

According to the literature [25] and the algorithm stability principle, the higher the ratio of task-relevant features learned by the model, the better its generalization. In this paper, for the image classification task, task-relevant features are defined as the features that are distributed over the category-specific object's location.

Figure 2 illustrates the feature distribution learned by the model and the annotated bounding boxes delineating the category object's location. The figure is organized into rows, with the top row depicting the input image processed by the model. The subsequent rows (second to fourth) correspond to three distinct models, each exhibiting varying levels of generalization capability. This research employed the concept of the generalization gap to quantify the models' generalization performance [1]. The generalization gap values, which are indicative of the model's generalization proficiency, are listed in the first column and were 4.0, 25.0, and 47.1, respectively, with the models' generalization ability diminishing from top to bottom. Upon examination of the figure, it is evident that the model with a robust generalization ability exhibited a high degree of overlap between the learned feature distribution and the object's location. The observed trends are characterized as follows: as the generalization ability diminishes, the learned feature distribution evolves from being concentrated and encompassing the entire target object, to a pattern where some features are dispersed outside of the object or are localized to specific areas within it, culminating in a scenario where the majority of features are localized outside the target object.

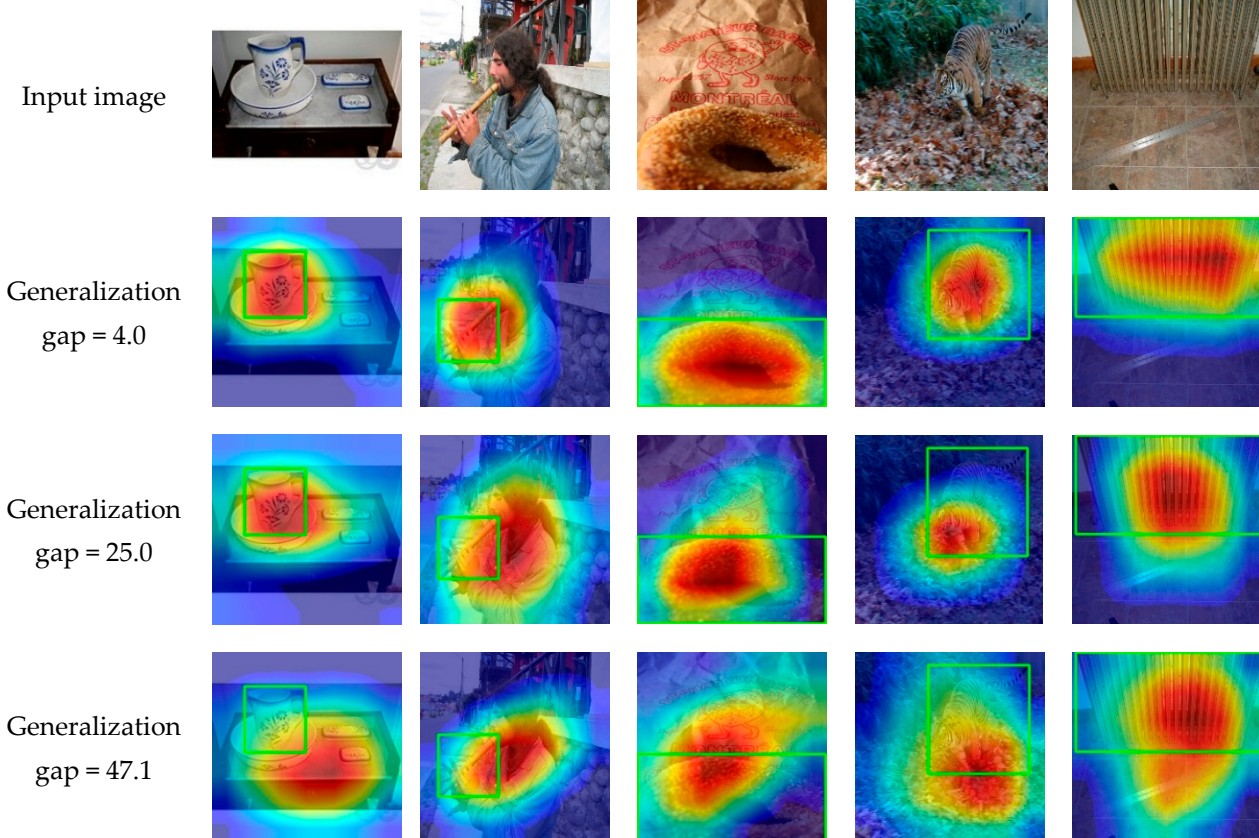

**Figure 2.** Examples of distribution of features learned by models and category object locations (i.e., the green bounding box) with different generalization abilities.

### 3.2. IoU-Based Generalizability Evaluation Methodology

Building upon the analyses presented in the preceding section, we identified a robust correlation between the spatial distribution of features within a model and its capacity for generalization. Consistent with our earlier observations, an increased alignment between the model's learned features and the spatial location of the category object within an image correlates positively with enhanced generalization performance, as indicated by a reduced generalization gap. In this research, we adopted the IoU metric to quantitatively evaluate the generalization capability of CNNs for image classification tasks, focusing on the ratio of task-relevant features.

As illustrated in Figure 3, our methodology commences with the extraction of the positional distribution of the category-specific object, delineated by its bounding box. Subsequently, we generate class activation maps that illustrate the spatial emphasis of features within the model's focus. The model's learned feature distribution is then delineated on these class activation maps, employing a threshold to segment and isolate the features that are most pertinent to the classification task. The penultimate stage uses IoU to calculate the ratio of task-relevant features, which quantifies the spatial overlap between the model's learned feature distribution and the actual positions of the category objects. This IoU value serves as a proxy for the proportion of accurately localized, task-relevant features, thereby offering a metric for the model's proficiency in generalizing from the training dataset to novel scenarios.

The above computational process can be formally defined as

$$\text{IoU}^{\text{Img}} = \frac{\sum \boldsymbol{A}_L \cap \boldsymbol{A}_G}{\sum \boldsymbol{A}_L \cup \boldsymbol{A}_G} \tag{2}$$

where $A_L$ and $A_G$, respectively, represent the feature distribution learned by the network and the position distribution of the category object. In the formula, $A_L$ is the portion of the feature distribution that has a weight greater than the threshold value, and $A_G$ is the rectangular bounding box of the corresponding category object in the ImageNet [17], and are, respectively, defined as follows:

$$A_L(i,j) = I(M^c(i,j), \rho) \tag{3}$$

$$I(x, \rho) = \begin{cases} 1 \text{ if } x \geq \rho \\ 0 \text{ if } x < \rho \end{cases} \tag{4}$$

$$A_G(i,j) = \begin{cases} 1 & (i,j) \, Inside \, the \, bounding \, box \\ 0 & (i,j) \, Outside \, the \, target \, box \end{cases} \tag{5}$$

where $M^c \in \mathbb{R}^{h \times w}$ represents the class activation map of the model relative to category c. Category c is the model's classification result of the input image. The value of each element in $M^c$ is normalized to [0, 1]. In the experiments in this study, the threshold was 0.1. The value range of $\text{IoU}^{\text{Img}}$ was [0, 1]. The larger the value, the higher the overlap between the feature distribution learned by the model and the location of the corresponding category object, which means a better the generalization ability of the model.

$A_L$ in Equation (3) is an irregular region similar to a circle, and we can similarly use its minimum outer rectangular box to calculate the IoU. In this paper, the calculation using the minimum outer rectangle is termed IoU-B. In our experiments, we also compared the performance of these two calculations.

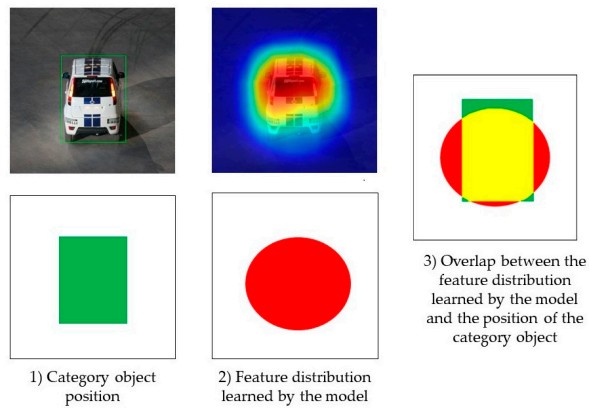

1) Category object position  2) Feature distribution learned by the model  3) Overlap between the feature distribution learned by the model and the position of the category object

**Figure 3.** Detailed flowchart for calculating the ratio of task-relevant features.

The IoU value calculated from a single image lacks statistical significance. In actual tasks, the $\text{IoU}^{\text{Model}}$ value at the model level is used as an indicator to judge the generalization ability of the model. $\text{IoU}^{\text{Model}}$ is defined as the mean IoU value calculated on all training images, that is,

$$\text{IoU}^{\text{Model}} = \frac{1}{N} \sum_i^N \text{IoU}_i^{\text{Img}} \tag{6}$$

where $N$ is the training set size, and $\text{IoU}^{\text{Img}}$ is the IoU value calculated from the $i$-th data.

In this section, we first discuss the CAM methods, which can be used to calculate the weights of features learned by the model at different pixels in the input image. Based on algorithm stability theory and qualitatively analyzing the relationship between the distribution of features learned by the model and the location of objects, we believe that the greater the feature weight learned by the model at the location of the category object, the stronger its generalization ability. Therefore, we used IoU to evaluate model generalization by calculating the degree of overlap between the feature distribution learned by the model and the location of the category object in the image.

## 4. Model Set Construction

### 4.1. Construction Method

To verify the effectiveness and accuracy of evaluating model generalization methods, a comprehensive model set is needed. The two existing model sets for the generalizability study were constructed on two simplified datasets, CIFAR and SVHN [1,3], which contain 756 and more than 10,000 CNN models, respectively. On the one hand, the image resolution in the CIFAR and SVHN datasets is only $32 \times 32$ pixels, with low data dimensions. On the other hand, the models in these sets mostly consist of around 10 layers with a small scale, which has a gap with the practical application scenarios. To bridge this gap and achieve a more precise and exhaustive validation of generalization assessment methodologies, this study constructed a model set with more than 2000 CNN models trained on the ImageNet subset by systematically varying commonly used hyperparameters. The models were developed utilizing the ResNet framework, specifically employing ResNet18, ResNet34, ResNet50, and ResNet101 architectures, which are prevalent in practical applications. The model set constructed in this study is comparable to the above two model sets in terms of quantity. Moreover, the size of the models is larger, and the images used for training are of higher resolution (with an average resolution of $496 \times 387$ pixels), aligning more closely with practical scenarios.

Taking into account the constraints of training duration, the study selected a random subset of 20 categories from the ImageNet dataset for experimental analysis. The data encompass a diverse array of categories, including animals, everyday objects, musical instruments, food items, and transportation-related items. This diversity is further accentuated by the extensive variation in the sizes and shapes of the objects represented within the dataset. A detailed breakdown of the data categories and their respective quantities used for model training is delineated in Table 1. For the purpose of determining test accuracy, a sample of 50 images per category was utilized, culminating in a comprehensive evaluation across 1000 images.

**Table 1.** Data categories and quantity information used for model training.

| Category | Number | Category | Number |
|---|---|---|---|
| Hair slide | 579 | Sleeping bag | 486 |
| German short-haired pointer | 493 | Koala | 560 |
| Eel | 500 | Tree frog | 683 |
| Brassiere | 487 | Tibetan mastiff | 859 |
| Ostrich | 512 | Gordon setter | 925 |
| Radiator | 450 | Standard schnauzer | 450 |
| Flute | 533 | Green mamba | 472 |
| Tiger | 542 | Grey whale | 421 |
| Cockroach | 562 | Bagel | 545 |
| Speedboat | 504 | Water jug | 610 |
| **Total** | | | **11,173** |

In the course of model training, we meticulously fine-tuned a suite of hyperparameters to generate a diverse array of CNN models. These hyperparameters, which encompass batch size, learning rate, optimization algorithm, regularization coefficients (weight decay), model architecture, data augmentation, and the utilization of pre-trained weights from the ResNet models on the ImageNet dataset, are widely recognized for their influence on the generalization capacity of machine learning models. Through an exhaustive exploration of various hyperparameter combinations, we successfully cultivated a spectrum of models with distinct generalization capabilities.

The hyperparameters for tuning can be formally defined as $p_i$, taking values from the set $\Theta_i$, for i = 1, …, $n$, and $n$ denoting the total number of hyperparameter types. In our study, 7 hyperparameters were selected, so $n = 7$. The selected hyperparameters were as follows:

1. **Batch size**: It determines the amount of data the model sees at each training step, which can influence the stability and diversity of the learning updates, thereby impacting the model's exposure to the overall data distribution. We hoped to unify the batch size used for training models with different network architectures, but we were limited by computing resources, so the maximum batch size we used for training was 64. For the sake of experimental diversity, we also chose values of 32 and 16 by dividing by 2;

2. **Learning rate**: It controls the step size the model takes during optimization, with larger rates potentially causing the model to overshoot minima and smaller rates leading to slower convergence, both of which can impact the model's ability to find a good balance between bias and variance. In our previous experiments, we found that when the learning rate was $1.25 \times 10^{-3}$, the model could be trained to convergence quickly, so we enlarged and reduced the learning rate by 100 times to $1.25 \times 10^{-1}$ and $1.25 \times 10^{-5}$ respectively to affect the training process of the model. This results in a model set with diverse generalization capabilities;

3. **Optimization algorithm**: It determines the path the model takes to minimize the loss function, which can influence the convergence speed and the quality of the solution found, thereby impacting the model's capacity to learn from the training data without overfitting. We chose two of the most common and more basic optimization algorithms: SGD and Adam;

4. **Regularization coefficient**: It controls the balance between fitting the training data and maintaining model simplicity, which helps prevent overfitting and encourages the model to learn more generalizable patterns. We empirically chose regular term coefficients $2 \times 10^{-4}$ and $5 \times 10^{-4}$, and then added 0, i.e., no regular term;

5. **Model structure**: It determines the complexity and representational capacity of the model, which directly influences its capability to capture underlying patterns without overfitting to the training data. Considering the computational resources and training time, the largest model structure we chose was ResNet101, and we also chose ResNet18, ResNet34, and ResNet50;

6. **Data augmentation**: It increases the diversity of the training data, which helps the model learn more robust features that can better represent the underlying data distribution, thus improving its performance on unseen data. Data augmentation is a common way to enhance the generalization ability of a model during training, so we chose to use or not apply data augmentation for training. In the experiments in this study, data enhancement was performed by randomly adding image flipping and color enhancement;

7. **Pre-trained weights**: They provide a good starting point with learned features from a vast dataset, which can transfer useful knowledge to new tasks, thereby reducing the need to learn from scratch and potentially improving the model's performance on similar data distributions. In our experiments, we found that using ResNet pre-training weights on ImageNet significantly affects the training time and generalization ability of the model, so we chose to use or not apply the pre-training weights.

Table 2 delineates the specific values assigned to each hyperparameter. For each value of hyperparameters, $\boldsymbol{p} \triangleq (p_1, \ldots, p_n) \in \Theta$, where $\Theta \triangleq \Theta_1 \times \ldots \times \Theta_n$.

In order to ensure that the model is trained to convergence, we established specific stopping conditions. These conditions were as follows:

1. The training loss function value should be below a threshold, which was set at 0.1. We used the cross-entropy loss function for this evaluation.
2. The model's accuracy on the training set needs to exceed a threshold of 0.95.
3. The loss function value decreases for two consecutive batches while the test error rate increases, which indicates overfitting and signals the need to stop training.
4. The training process should not exceed 150 epochs. This condition ensures that the training can be completed within a limited time frame. For more implementation details, see Appendix A.

The first two termination conditions are to ensure that the model can converge, and the last two conditions ensure that training ends within a limited period. We saved one or two models for each set of training parameters. The training process stops when any two of the first three conditions are met or directly when the fourth condition is met. We repeated the experiment twice for each set of parameters. In theory, $|\Theta| \times 2 \times 2 = 3456$ models can be generated, but taking into account the time factor, we utilized four GTX 2080ti GPUs and dedicated nearly 30 days to train a total of over 2100 models. Eventually, we selected 2000 models that achieved a training accuracy greater than 80% to comprise our model set. Within this set, 500 models were chosen for each network structure. The reason for this choice is that the training of these models is near convergence, which aligns well with the actual application scenario. Only a small number of models did not converge, for example, the training accuracy of the model after training 150 batches was only 20%.

**Table 2.** Hyperparameters used in model training.

| Hyperparameter | Value Range |
|---|---|
| Batch size | $\{16, 32, 64\}$ |
| Learning rate | $\{1.25 \times 10^{-1},\ 1.25 \times 10^{-3},\ 1.25 \times 10^{-5}\}$ |
| Optimization algorithm | $\{\text{Adam, SGD}\}$ |
| Regularization coefficient | $\{0,\ 2 \times 10^{-4},\ 5 \times 10^{-4}\}$ |
| Model structure | $\{\text{ResNet18, ResNet34, ResNet50, ResNet101}\}$ |
| Data augmentation | $\{\text{True, False}\}$ |
| Pre-trained weights | $\{\text{True, False}\}$ |

*4.2. Model Distribution*

The construction of a high-quality dataset is fundamental to the investigation of methodologies for assessing model generalization capabilities. The distribution of these capabilities is a critical metric for evaluating the dataset's quality.

A box plot, a versatile graphical tool, provides a comprehensive statistical summary of the data distribution, delineating critical aspects such as central tendency and dispersion. Specifically, it captures the median; the interquartile range (IQR), which includes the central 50% of the data; and the lower (Q1) and upper (Q3) quartiles corresponding to the 25th and 75th percentiles, respectively. The plot also highlights potential outliers, represented as individual points beyond the whiskers that extend to 1.5 times the IQR from the quartiles. Collectively, these elements offer a succinct yet informative overview of the variability and central tendency of the generalization gap values across the spectrum of model architectures within the established dataset.

Figure 4 illustrates the distribution of generalization capabilities across various structural models within the curated dataset. The generalization gap was employed to quantify the generalization ability of each model. Upon scrutiny of Figure 4, it becomes evident that the generalization gap values exhibited a wide range, signifying considerable heterogeneity in the generalization capabilities of the constituent models. Furthermore, the consistency in the distribution of generalization capabilities across different model architectures precludes the confounding effects of structural variance on the validation of the generalization assessment methodology.

Figure 5 illustrates the quantity distribution of the models across various ranges of generalization gaps. The results depicted in the figure reveal that the majority of models were concentrated within the extremes of very small or very large generalization gaps, while only a few models exhibited intermediate values. This distribution pattern can be attributed to the training termination conditions employed during the model training process, which aimed to achieve convergence. Converged models tend to demonstrate either minimal or substantial generalization gaps. A small generalization gap indicates proficient generalization to unseen data, whereas a large gap suggests overfitting, wherein the model incorporates task-irrelevant features from the training data, resulting in impressive performance on the training set but notable degradation on new data. Models displaying

intermediate generalization gap values are in a transitional state, indicating that they have not yet attained the optimal balance between bias and variance. Such models may benefit from additional training. In summary, the quantity distribution of models across different generalization gap ranges, as depicted in Figure 5, signifies that the majority of models in the dataset had reached a state of convergence. This achievement serves as a foundation for the subsequent methodological investigations into assessing model generalization capability and validating the methodology's efficacy. Furthermore, this distribution validates the efficacy of the established training termination conditions, ensuring both the quality and efficiency of model training.

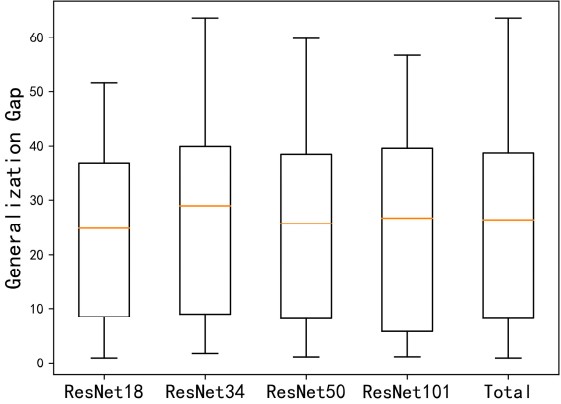

**Figure 4.** Generalization gap distribution in the model set.

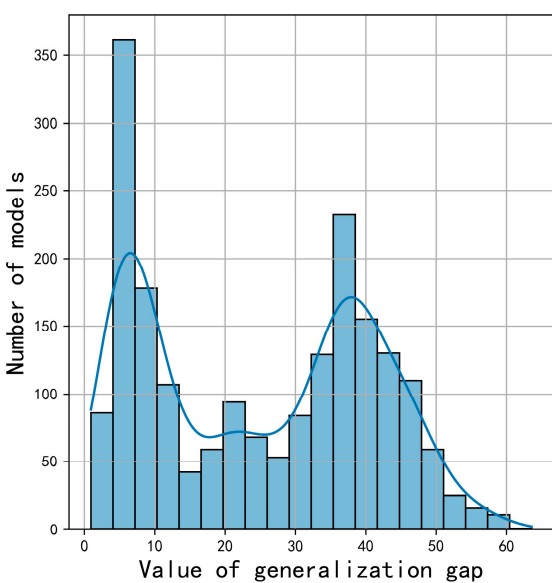

**Figure 5.** Quantity distribution of models in different generalization gap ranges.

In conclusion, this chapter introduced a large-scale model set that addresses the limitations of existing sets. The diverse range of generalization abilities and consistent distribution across different model structures make this model set valuable for studying and validating generalizability methods.

## 5. Experimental Analysis

### 5.1. Evaluation Metric

In this study, the Pearson correlation coefficient was selected as the criterion to check the effectiveness of using IoU to evaluate model generalization capabilities. The correlation between IoU and the model's generalization ability, which is also referred to as the generalization gap, was used to assess the effectiveness of the method. The Pearson correlation

coefficient is a widely used measure of linear correlation between two sets of data. It is calculated as the ratio of the covariance of the two variables to the product of their standard deviations and ranges from −1 to 1. A correlation coefficient closer to 1 or −1 indicates a stronger correlation between the two variables.

$$r_{xy} = \frac{\sum_{i=1}^{n} (x_i - \overline{x})(y_i - \overline{y})}{\sqrt{\sum_{i=1}^{n} (x_i - \overline{x})^2} \sqrt{\sum_{i=1}^{n} (y_i - \overline{y})^2}} \tag{7}$$

where $n$ represents the sample size, and $\overline{x}$ and $\overline{y}$ are the mean values of random variables **x** and **y**, respectively. In the following experiment, the main objective was to calculate the correlation coefficient between the IoU and the model's generalization gap. If there is a significant correlation between these two factors, it demonstrates the effectiveness of the evaluation method. A strong correlation would indicate a high accuracy of the evaluation approach.

*5.2. Experimental Results*

5.2.1. Comparison Experiments

In this section, we used the Pearson's correlation coefficient of the constructed model set to verify the validity of the model generalizability assessment approach. This rigorous analysis ensures the effectiveness and accuracy of the assessment methodology.

The IoU values of the feature distribution and object position for all models in the proposed model set, as well as the generalization gaps of each model, were computed in this study. For calculating the class activation maps, the Grad-CAM method implemented in Captum [26] was used. In this study, three comparison methods were selected, namely Spectral Norm [8], Nuclear Norm [21], and Effective Invariance (EI) [22]. Since the original Spectral Norm method cannot be directly applied to CNN models, the implementation approach described in reference [3] was used. The details of these methods can be found in Appendix B.

Table 3 delineates the correlation coefficients between the generalization assessment methods and generalization gap across various ResNet architectures. Each row corresponds to a distinct ResNet architecture, with the final row summarizing the results across all the models. The columns indicate the model subsets with training accuracies exceeding specific thresholds. Ideally, all methods should correlate positively with a model's generalization capacity, which is inversely proportional to its generalization gap. Consequently, a negative correlation between the methods and generalization gaps is anticipated. The effectiveness of an assessment method is indicated by a smaller absolute correlation coefficient value. The most promising results within each category are emphasized in bold.

The analysis from Table 3 reveals a strong correlation (absolute value > 0.8) between IoU and the IoU-B method and the generalization gap, e.g., the model's generalization capacity. Due to the potential noise introduced by using the minimum bounding box in the IoU-B calculations, the correlation between IoU-B and the generalization gap was weaker than that of IoU. Spectral Norm had almost no correlation with the generalization ability of the model, which means that this method is not suitable for generalizability assessments of deep neural network models. Nuclear Norm was the least computationally intensive and most efficient, but its correlation with the generalization ability of the model was weak. The correlation between EI and the generalization ability of the model was second only to the IoU-based method, but its computation was computationally intensive as it needs to process the input and its multiple transformed versions. In summary, the evaluation method in this paper using IoU to calculate the ratio of task-relevant features had the highest accuracy and is computationally efficient.

As training accuracy thresholds rise, the correlation between the evaluation methods and model generalization is strengthened. This trend arises because lowering the threshold broadens the spectrum of model accuracies without altering the generalization ability range, thus complicating model differentiation. For instance, models with training accu-

racies of 96% and 86%, and respective test accuracies of 93% and 83%, yielded identical generalization gaps of 3%. However, achieving the same IoU value for these models is implausible due to the inherent differences in their class activation maps.

**Table 3.** Correlation coefficients between different valuation methods and the model generalization gap.

| Accuracy Threshold | | $\geq$**95** | $\geq$**90** | $\geq$**85** | $\geq$**80** |
|---|---|---|---|---|---|
| ResNet18 | Spectral Norm | 0.0592 | 0.0330 | 0.0429 | 0.0451 |
| | Nuclear Norm | $-0.3243$ | $-0.2874$ | $-0.2793$ | $-0.2273$ |
| | EI | $-0.6577$ | $-0.6550$ | $-0.6539$ | $-0.6462$ |
| | IoU-B | $-0.8587$ | $-0.8556$ | $-0.8466$ | $-0.8358$ |
| | IoU | $\mathbf{-0.9144}$ | $\mathbf{-0.9096}$ | $\mathbf{-0.8996}$ | $\mathbf{-0.8911}$ |
| ResNet34 | Spectral Norm | 0.0579 | 0.0324 | 0.0421 | 0.0424 |
| | Nuclear Norm | $-0.4963$ | $-0.4934$ | $-0.4695$ | $-0.4432$ |
| | EI | $-0.7316$ | $-0.7339$ | $-0.7324$ | $-0.7303$ |
| | IoU-B | $-0.8207$ | $-0.8205$ | $-0.8139$ | $-0.8085$ |
| | IoU | $\mathbf{-0.8273}$ | $\mathbf{-0.8252}$ | $\mathbf{-0.8183}$ | $\mathbf{-0.8149}$ |
| ResNet50 | Spectral Norm | 0.0574 | 0.0322 | 0.0416 | 0.0410 |
| | Nuclear Norm | $-0.5610$ | $-0.5478$ | $-0.5063$ | $-0.4649$ |
| | EI | $-0.7676$ | $-0.7638$ | $-0.7642$ | $-0.7614$ |
| | IoU-B | $-0.8140$ | $\mathbf{-0.8135}$ | $\mathbf{-0.8131}$ | $-0.8040$ |
| | IoU | $\mathbf{-0.8171}$ | $-0.8129$ | $-0.8125$ | $\mathbf{-0.8054}$ |
| ResNet101 | Spectral Norm | 0.0574 | 0.0323 | 0.0417 | 0.0401 |
| | Nuclear Norm | $-0.5707$ | $-0.5599$ | $-0.5211$ | $-0.489$ |
| | EI | $-0.7963$ | $-0.8021$ | $-0.8056$ | $-0.8076$ |
| | IoU-B | $-0.8426$ | $-0.8417$ | $-0.8402$ | $-0.8399$ |
| | IoU | $\mathbf{-0.8571}$ | $\mathbf{-0.8540}$ | $\mathbf{-0.8523}$ | $\mathbf{-0.8520}$ |
| Total | Spectral Norm | 0.0204 | 0.0198 | 0.0195 | 0.0195 |
| | Nuclear Norm | $-0.4668$ | $-0.4554$ | $-0.4357$ | $-0.3898$ |
| | EI | $-0.7413$ | $-0.7419$ | $-0.7425$ | $-0.7398$ |
| | IoU-B | $-0.8110$ | $-0.8116$ | $-0.8070$ | $-0.8005$ |
| | IoU | $\mathbf{-0.8248}$ | $\mathbf{-0.8228}$ | $\mathbf{-0.8200}$ | $\mathbf{-0.8159}$ |

Among models with varying architectures, the IoU's correlation with generalization ability was consistent. Notably, the ResNet18 model demonstrated significantly higher correlation coefficients, potentially attributable to its simpler architecture and enhanced class activation map generation. The ResNet101 model also showed slightly higher correlation coefficients than the ResNet34/50 models, which may be influenced by the limited sample size and associated randomness in the training process.

As the model layer depth increased, the correlation coefficient of IoU and the generalization gap slightly diminished, whereas the EI correlation coefficient exhibited a slight rise. This trend is attributed to the use of a dataset with merely 20 categories in this study, contrasting with EI's typical application in models with around 1000 categories. The limited category scope represents a constraint in this study.

5.2.2. Model Selection Experiments

The generalizability evaluation method can be used for model selection. Given two models $\text{Model}_1$ and $\text{Model}_2$, the IoU values of the two models are $\text{IoU}_1^{\text{Model}}$ and $\text{IoU}_2^{\text{Model}}$. If $\text{IoU}_1^{\text{Model}} > \text{IoU}_2^{\text{Model}}$, then $\text{Model}_1$ is considered to have a stronger generalization ability than $\text{Model}_2$, and accordingly, the model with a stronger generalization ability can be selected. In the model set, pairwise combinations were made, and the IoU was used to select models with stronger generalization abilities. Then, the correctness of the selection was counted.

Table 4 presents the results of the model selection experiments, and compares the model selections using IoU within identical network structures (first four rows) and across

different structures (last row). The high accuracy rate of the IoU-based model selection underscores the method's effectiveness and feasibility. Despite the significant impact of training data quality on model generalization, the IoU metric, calculated using the model's predicted category, remained robust even with low-quality, noisy data. This suggests that the IoU can effectively identify models with superior generalization capabilities, even when training data are suboptimal.

**Table 4.** The accuracy of using IoU to select models with strong generalization ability.

| Accuracy Threshold | ≥95 | ≥90 | ≥85 | ≥80 |
|---|---|---|---|---|
| ResNet18 | 76.10% | 75.74% | 75.14% | 74.67% |
| ResNet34 | 73.81% | 73.25% | 72.83% | 72.57% |
| ResNet50 | 72.78% | 72.30% | 72.01% | 71.71% |
| ResNet101 | 76.67% | 76.28% | 76.00% | 75.77% |
| Total | 74.36% | 74.18% | 73.97% | 73.82% |

*5.3. Validity Testing*

In this section, a comparison experiment was performed to calculate the correlation coefficients between the IoU value of the trained models and the randomly initialized models and their generalization ability, respectively. The weight values of the random models were all randomly initialized. For the ResNet18, ResNet34, ResNet50, and ResNet101 network architectures, 50 random models were generated for each, resulting in a total of 200 random models. The trained models were obtained using the model set constructed in this study.

Table 5 reveals a robust correlation between the IoU values and the generalization capabilities of trained models. In contrast, the correlation between the IoU values for randomly initialized models and their generalization was negligible, resembling random chance. This comparative analysis validates the use of IoU as a reliable metric for assessing the generalizability of ResNet models.

**Table 5.** Comparison of the correlation between IoU and model generalization ability between trained and randomly initialized models.

| | ResNet18 | ResNet34 | ResNet50 | ResNet101 | Total |
|---|---|---|---|---|---|
| Random Model | −0.1544 | 0.1820 | 0.2246 | −0.0585 | −0.0094 |
| Trained Model | −0.8911 | −0.8149 | −0.8054 | −0.8520 | −0.8159 |

*5.4. Robustness Testing*

5.4.1. CAM Method Replacement Testing

To thoroughly validate the evaluation approach by calculating the ratio of task-relevant features using IoU values, we used GradCAM++ [27] and SmoothGradCAM++ [28] to recalculate the IoU of the ResNet18 models, and calculated the correlation between IoU and the generalization gap.

Table 6 displays the experimental findings, highlighting a consistent strong correlation between computed IoU values and the generalization gap across different CAM calculation methods, including GradCAM, GradCAM++, and SmoothGradCAM++. This suggests that the IoU-based evaluation approach is largely invariant to the CAM method chosen, underscoring its robustness in assessing model generalization.

**Table 6.** The correlation between IoU and generalization gap based on using different CAM calculation methods on the ResNet18 models.

| Accuracy Threshold | ≥95 | ≥90 | ≥85 | ≥80 |
|---|---|---|---|---|
| IoU (Using GradCAM++) | −0.9127 | −0.9105 | −0.9004 | −0.8879 |
| IoU (Using SmoothGradCAM++) | −0.8872 | −0.8873 | −0.8731 | −0.8582 |
| IoU (Using GradCAM) | −0.9144 | −0.9096 | −0.8996 | −0.8911 |

### 5.4.2. Bounding Box Replacement Testing

To ease the manual bounding box annotation efforts, we employed a saliency detection-based approach to generate pseudo-labels, thereby eliminating the need for manual editing. Specifically, we utilized the MCCL algorithm [29] to produce a saliency map, which was subsequently binarized to create a mask for IoU (Intersection over Union) calculations. To calculate the IoU in this case, we denoted the non-zero portion of the saliency map as $A_G$ in Equation (4). Figure 6 illustrates the method's effectiveness through a comparison of the original image, labeled frames, and the corresponding saliency map.

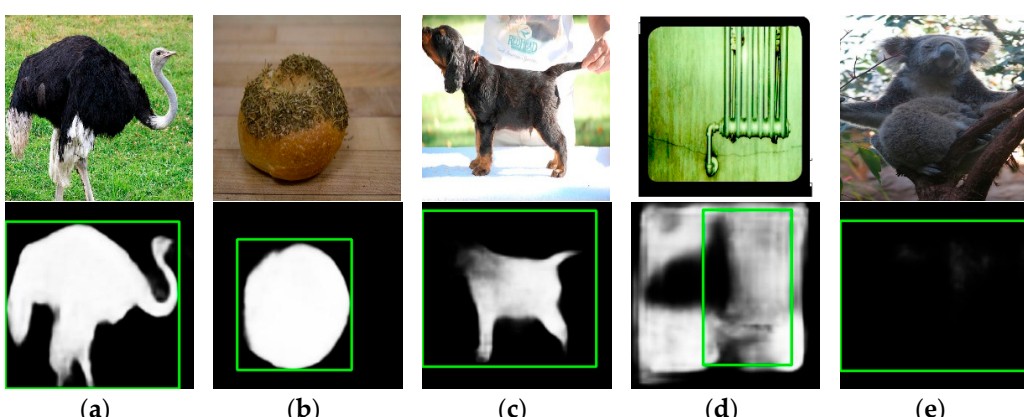

(a)　　(b)　　(c)　　(d)　　(e)

**Figure 6.** Examples of generated saliency maps and original manually labeled bounding boxes (i.e., the green box), where the first row is the original image and the second row is the corresponding saliency map and bounding box. Columns (**a**–**e**) are five examples, in descending order of quality from the saliency map.

Table 7 presents the correlation between generalization gap and IoU derived from the manually labeled bounding boxes and automatically generated saliency maps for the ResNet18 models. The findings indicate that a robust correlation persisted between generalization gap and IoU, albeit slightly diminished when using saliency maps compared to manual bounding boxes. This attenuation is attributed to the inherent limitations of saliency maps in precisely delineating the spatial extent of the subject category. As illustrated in Figure 6, while columns a and b exhibited accurate saliency mapping, the precision deteriorated across columns c, d, and e, culminating in column e's failure to identify the objects within the image.

From the results of this experiment, it can be seen that the use of pseudo-labels (saliency maps) to calculate the ratio of task-relevant features is still valid. This alleviates the dependence on manually labeled bounding boxes.

**Table 7.** The correlation between generalization gap and IoU calculated based on manually labeled bounding boxes and automatically generated saliency maps from the ResNet18 models.

| Accuracy Threshold | ≥95 | ≥90 | ≥85 | ≥80 |
|---|---|---|---|---|
| IoU (using saliency map) | −0.8465 | −0.8425 | −0.8322 | −0.8215 |
| IoU (using bounding box) | −0.9144 | −0.9096 | −0.8996 | −0.8911 |

## 6. Conclusions

This study presents a new perspective on evaluating the generalization ability of CNN models by using IoU, which addresses the issue of poor accuracy and applicability of CNN model generalization evaluation methods for the task of image classification. The method, based on the ResNet architecture, leverages IoU to quantify the ratio of task-relevant features within the model's learned feature set. Task-relevant features are identified through Class Activation Maps (CAMs), focusing on the object's location within

an image. To validate our method, we trained 2000 ResNet models on a subset of ImageNet, providing a robust dataset for generalization analyses.

Our findings underscore the method's effectiveness and accuracy, with the IoU calculations only requiring training data and a trained model, thereby enhancing computational efficiency. This approach not only evaluates model generalization but also aids in selecting optimal models and tuning hyperparameters. The method's robustness was evidenced by its minimal sensitivity to variations in CAM algorithms. However, it should be noted that the reliance on bounding box annotations for the IoU calculation presents a limitation.

In conclusion, the IoU-based method offers a promising framework for evaluating CNN model generalization in image classification tasks. The extensive model set developed in this study lays a solid foundation for future research. While current practices necessitate manual bounding box annotations, future work will explore automated annotation techniques to address this limitation.

**Author Contributions:** Conceptualization, Q.Z.; methodology, A.D.; software, A.D.; validation, A.D. and Y.D.; resources, Q.Z.; data curation, A.D.; writing—original draft preparation, A.D.; writing—review and editing, Y.D.; supervision, Q.Z.; All authors have read and agreed to the published version of the manuscript.

**Funding:** This research received no external funding.

**Institutional Review Board Statement:** Not applicable.

**Informed Consent Statement:** Not applicable.

**Data Availability Statement:** Data and code will be available at https://github.com/AndyDu0116/generalizationEvaluate, accessed on 29 April 2024.

**Conflicts of Interest:** The authors declare no conflicts of interest.

## Appendix A

To ensure that the models are trained within a feasible timeframe, we established a cap on the number of training epochs. Our experimental observations revealed that 90% of the models met the termination criteria within the initial 50 epochs. Consequently, we set the upper limit at 150 epochs, which is threefold higher than the base threshold. We performed a detailed statistical analysis on the training loss and accuracy for models terminated at the 150-epoch mark. As depicted in Figure A1, the loss and accuracy curves for a selection of models trained for 150 epochs exhibited a stabilizing trend over the last 20 to 10 epochs. Moreover, we computed the relative change in loss and accuracy over the final 10 epochs relative to the entire training duration for these models. The findings suggest that the variation in both loss and accuracy during the last 10 epochs is minimal (less than 1%). These outcomes confirm that a 150-epoch training regime is adequate to ensure the thorough completion of the training process.

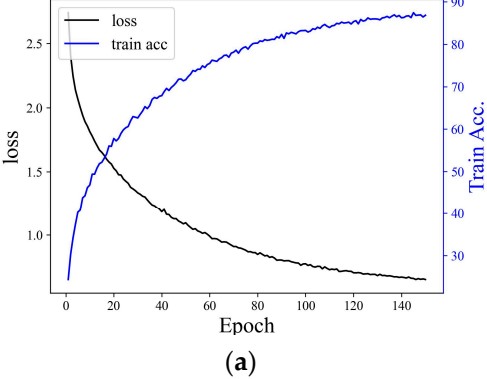
(a)

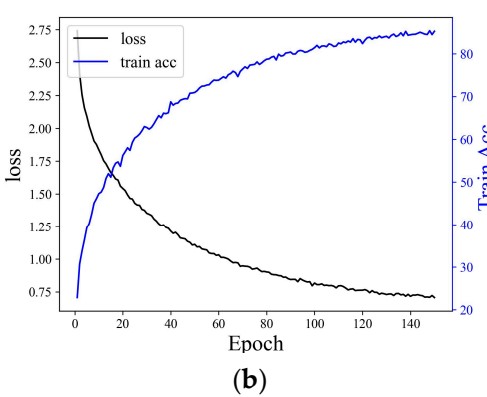
(b)

**Figure A1.** *Cont.*

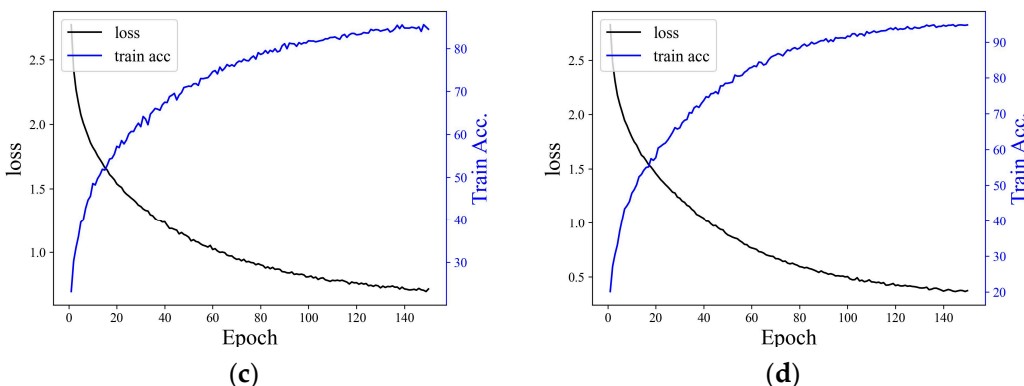

**(c)**                                              **(d)**

**Figure A1.** The curves illustrate the variation in loss and accuracy during the training of a model trained for 150 epochs. (**a**–**d**) Training process of four models under different hyperparameters.

### Appendix B

Below, we briefly describe the generalization evaluation methods compared in our work. We denote the model as $f$.

**Spectral Norm [8].** This method evaluates the generalizability of the model using the product of the spectral norm of the layer weights multiplied by the sum over layers of the ratio of Frobenius norm to the spectral norm of the layer weights [3].

$$\text{SpectralNorm}(f) = \frac{\prod_{i=1}^{d} \|W_i\|_2^2 \sum_{j=1}^{d} \frac{\|W_j\|_F^2}{\|W_j\|_2^2}}{\gamma^2} \tag{A1}$$

where d is the number of layers in the model $f$. $W$ is the weight matrix of the model. $\|W\|_2^2$ denotes the square of the spectral norm of the tensor $W$, and $\|W\|_F^2$ denotes the square of Frobenius norm of the tensor $W$. $\gamma$ is the 10th percentile of the margin values on the training set.

**Nuclear Norm [21].** This method uses the normalized value of the nuclear norm of the prediction matrix to evaluate the generalization ability of a model.

$$\text{NuclearNorm}(f) = \frac{1}{N} \left[ \frac{\|P\|_*}{\sqrt{\min(N, K) \cdot N}} \right] \tag{A2}$$

where N is the number of samples in the training set, and K is the number of classes. $\|P\|_*$ is the nuclear norm of $P$. $P \in \mathbb{R}^{N \times K}$ denotes the prediction matrix of $f$ on the training set, and $P_{i,:}$ is the softmax vector of $i$-th sample (*i.e.*, $P_{i,:} = softmax(f(\mathbf{x}_i))$).

**Effective Invariance (EI) [22].** This method evaluates the generalization ability of a model by quantifying its invariance to transformations of the input data.

$$\text{EI}(\mathbf{x}, \text{T}(\mathbf{x}), f) = \begin{cases} \sqrt{\hat{p}_t \cdot \hat{p}} & \text{if } \hat{y}_t = \hat{y}; \\ 0 & \text{otherwise.} \end{cases} \tag{A3}$$

where $\mathbf{x}$ is the input sample in the training set. $\text{T}(\mathbf{x})$ represents the transformation of $\mathbf{x}$, including 3 rotationally transformed angles ($90°$, $180°$, and $270°$). The EI value is the average of the values calculated for the three rotation angles. $\hat{p}_t$ is the predicted softmax vector of the model for the transformed input sample, and $\hat{p}$ is the predicted output of the model for the original input sample. $\hat{y}_t$ and $\hat{y}$ are the model's classification results for the transformed and original input samples, respectively.

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
