# Peer review of "Methodology for Evaluating the Generalization of ResNet"

_applsci, doi:10.3390/app14093951_

Round 1

Reviewer 1 Report (Previous Reviewer 3)

Comments and Suggestions for Authors

modify the title where authors focused on resnet models do replace with that 

Also, do add the similar works for comparing contributions 

It is also encouraged to add the 20 class experiment details, then followed by the rest of the analysis w.r.t. that class or object limitation 

Comments on the Quality of English Language

Do update the starting statements for the paragraphs where all the tables and figures are discussed.

Author Response

Reviewer 2 Report (Previous Reviewer 2)

Comments and Suggestions for Authors

This paper gives some insights of the characteristics of IoU when applied to CNN. As pointed out in previous works, IoU well works in evaluating the accuracy of CNN. In this paper, some quantitative evaluation is added to further charcterize IoU.

Comments on the Quality of English Language

Usually, the tense of a paper is "present."

Author Response

Dear Reviewer,

Thank you very much for taking the time to review this manuscript. We greatly appreciate your valuable feedback.

We have made modifications to the tenses throughout the entire article. Except for the statements regarding the existing previous work, we have revised the other parts to the present tense. Please find the detailed responses below and the corresponding revisions highlighted in the re-submitted files.

Best regards

Reviewer 3 Report (Previous Reviewer 1)

Comments and Suggestions for Authors

The Authors have sligthly modified the submission, comparing to the previous version of the manuscript, which has been rejected. In fact, I can't see the substantial quality improvement of the current  version of the submission, as the modification mostly consist in adding some paragraphs of the text and changing the order of some tables.  That's why for me it is still the borderline paper, of rather limited interest to readers.

And the submission still does not contain the formal problem statement - in my opinion the paragraph added to Section 3.2 doesn't fulfill such requirements.
Also the choice of parameters of simulation experiments (lines 274-305), which substantially influence the results, should be more carefully explained.

Author Response

This manuscript is a resubmission of an earlier submission. The following is a list of the peer review reports and author responses from that submission.

Round 1

Reviewer 1 Report

Comments and Suggestions for Authors

Thw introductory part (sections 1 and 2) is too long, it takes over a quarter of the text.

It is not easy to find a formal problem statement and a summary of original contribution of the Authors - these issues must be clearly addressed in the improved version of the submission.

The influence of parameter values (listed in Table 2 and below) on the results of the study is essential. So the Authors should carefully explained their choices.

The results presented in Fig. 6 should be discussed and the explanation of their meaning should be provided.

Similarly - there is a lack of interpretation, what does for the reader practically follow from the distributions presented in Fig. 5? In fact, those desults are not clear and impressive.

Reviewer 2 Report

Comments and Suggestions for Authors

This paper, titled "Methodology for Evaluating the Generalization of Convolutional Neural Network," proposes RFRM, a new evaluation metrics for CNN, which is built upon (Grad-)CAM.

In fact, RFRM is well known as IoU, one of standard metrics for computer vision, and its effectiveness is well examined.

I am afraid that I do not find a value additional to conventional IoU.

Comments on the Quality of English Language

none

Reviewer 3 Report

Comments and Suggestions for Authors

the proposed work on Evaluating the Generalization of Convolutional Neural Network is an impressive study: 

The study focus more on Resnet based models for assessment of learning generalizability. 

It is suggested to include more RFRM for scorecam, gradcamGradCAM, GradCAM++, Smooth Grad-CAM++ and Score CAM. with multiple objects in a frame. 

What is the scope of the proposed model for assessing the other similar models which were bench marked on  imagenet dataset 

However it is seen in recent days MobileNet and EfficientNet family based algorithms showed scaled and outperformed the existing methods. do provide a thought on that 

Reviewer 4 Report

Comments and Suggestions for Authors

- What the author sees as the main contribution of his work. The RFRM issue is extensive and discussed in many papers. The solution presented in the paper just looks like an increase in the quantity of trained models without adding any scientific contribution from the researchers.

- The quality of the training data is very important. How would the algorithm perform when the data quality is worse compared to classically used CNN models. Wouldn't the classical CNN models be more robust?

- How the authors intend to address the problem of manual editing of bounding box annotations in the future. With a high number, algorithmization and logic editing will necessarily be needed. Is the algorithm presented in this way applicable in practice?

Round 2

Reviewer 1 Report

Comments and Suggestions for Authors

The Authors tried to improve the previous version of the manuscript following the remarks of the reviewer. In this version the submission can be consudered for publication, however I am still not convinced about its interest to readers.

The issues which should be improved in the minorrevision of the submission:

-- Formal problem statement - in my opinion it is still not clearly described.

-- Figure 1 should be removed, as misleading. The test dataset is not used for additional training, but for evaluation of the performance of the network - atfer completion (!) of the training. The network is not trained on the test set, what is also mentioned by the Authors in line 40 ("test set when the model has converged"). So the curve illustraiting the accuracy vs. the number of training epochs has no sense.

-- Figure 3 - the word "Generalization" can't be wrapped.

-- In line 300 the Authors write: "The training process should not exceed 150 epochs". Can such a number guarantee the completion of the training process?

Reviewer 2 Report

Comments and Suggestions for Authors

This is the second round. I reviewed the response letter from the authors. I understand that the goal of this paper is evaluation of CNN by using RFRM.

However, I must say again that RFRM is the same as IoU, and it is originally coined to evaluate the accuracy of object detection based on CNN. We see a number of papers for the same methodology and the same goal.  By searching scholar.google.com for "IoU evaluation CNN", we can find 21600 references.

What the authors should have done is to show "RFRM is different to IoU." and "Using RFRM is better than using other methods including IoU." Using only spectral norm is totally insufficient. 

Comments on the Quality of English Language

none

Reviewer 4 Report

Comments and Suggestions for Authors

I am missing the incorporation of my comment number 1 (Somewhere in the introduction ) and 3 (Somewhere in the end) in the text. "What the author sees as the main contribution of his work. The RFRM issue is extensive and discussed in many papers. The solution presented in the paper just looks like an increase in the quantity of trained models without adding any scientific contribution from the researchers."

However, I still miss highlighting your contribution in the text because the method mentioned in the article using "Generalization evaluation method named RFRM (Relevant Feature Ratio Method) for CNN models based on Class Activation Map (CAM)" is not new, which can be seen in the year of citation of the method and its relatively common and frequent deployment in real practice.

I also miss the consideration in the text of the cavality of the training data and the implications for the outcome according to comment number 2. "The quality of the training data is very important. How would the algorithm perform when the data quality is worse compared to classically used CNN models. Wouldn't the classical CNN models be more robust?"

"How the authors intend to address the problem of manual editing of bounding box annotations in the future. With a high number, algorithmization and logic editing will necessarily be needed. Is the algorithm presented in this way applicable in practice?"

Round 3

Reviewer 2 Report

Comments and Suggestions for Authors

Seeing the response from the authors, I understand that the authors would like to find a certain metric based on (Grad-)CAM. I am afraid that the coined metric was IoU, or at best its minor variant. As you know, IoU is an established metric. Probable contribution of the authors would be an application of IoU to other fields than object detection. Then, the authors should have explicitly written "application of IoU to ..." instead of "proposal of RFRM."

From the response letter, the performance of RFRM is comparable to IoU1. I cannot find a major technical breakthrough there.

Furthermore, the content of response letter is not reflected in Table 3.

Comments on the Quality of English Language

none
